# Dietary Gamma-Aminobutyric Acid (GABA) Induces Satiation by Enhancing the Postprandial Activation of Vagal Afferent Nerves

**DOI:** 10.3390/nu14122492

**Published:** 2022-06-16

**Authors:** Utano Nakamura, Taichi Nohmi, Riho Sagane, Jun Hai, Kento Ohbayashi, Maiko Miyazaki, Atsushi Yamatsu, Mujo Kim, Yusaku Iwasaki

**Affiliations:** 1Department of Research and Development, Pharma Foods International Co., Ltd., 1-49 Goryo-Ohara, Nishikyo-ku, Kyoto 615-8245, Japan; u-nakamura@pharmafoods.co.jp (U.N.); r-sagane@pharmafoods.co.jp (R.S.); m-miyazaki@pharmafoods.co.jp (M.M.); a-yamatsu@pharmafoods.co.jp (A.Y.); mujokim@pharmafoods.co.jp (M.K.); 2Laboratory of Animal Science, Graduate School of Life and Environmental Sciences, Kyoto Prefectural University, 1-5 Hangi-cho, Shimogamo, Sakyo-ku, Kyoto 606-8522, Japan; s821631034@kpu.ac.jp (T.N.); s822731002@kpu.ac.jp (K.O.); 3Laboratory of Functional Anatomy, Faculty of Agriculture, Kyushu University, 744 Motooka, Nishi-ku, Fukuoka 819-0395, Japan; junhai.0422@gmail.com

**Keywords:** GABA, dietary GABA, vagal afferents, nodose ganglion, food intake, postprandial satiation, vagotomy, capsaicin-sensitive sensory nerves, ERK

## Abstract

Gamma-aminobutyric acid (GABA) is present in the mammalian brain as the main inhibitory neurotransmitter and in foods. It is widely used as a supplement that regulates brain function through stress-reducing and sleep-enhancing effects. However, its underlying mechanisms remain poorly understood, as it is reportedly unable to cross the blood–brain barrier. Here, we explored whether a single peroral administration of GABA affects feeding behavior as an evaluation of brain function and the involvement of vagal afferent nerves. Peroral GABA at 20 and 200 mg/kg immediately before refeeding suppressed short-term food intake without aversive behaviors in mice. However, GABA administration 30 min before refeeding demonstrated no effects. A rise in circulating GABA concentrations by the peroral administration of 200 mg/kg GABA was similar to that by the intraperitoneal injection of 20 mg/kg GABA, which did not alter feeding. The feeding suppression by peroral GABA was blunted by the denervation of vagal afferents. Unexpectedly, peroral GABA alone did not alter vagal afferent activities histologically. The coadministration of a liquid diet and GABA potentiated the postprandial activation of vagal afferents, thereby enhancing postprandial satiation. In conclusion, dietary GABA activates vagal afferents in collaboration with meals or meal-evoked factors and regulates brain function including feeding behavior.

## 1. Introduction

Gamma-aminobutyric acid (GABA), a four-carbon nonproteinogenic amino acid, is found in the mammalian brain [1] and serves as the main inhibitory neurotransmitter [2]. GABA is also naturally present in various foods, such as tea, soybean, germinated rice, some vegetables and fruits, and some fermented foods [3,4,5]; it is acquired from our daily normal diet. Recently, it has been manufactured and widely used as a dietary supplement. Previous studies in human trials demonstrated that dietary GABA induces antihypertensive effects [6], alleviates anxiety [7], and improves sleep quality [8] and cognitive functions [9,10]. As a result, GABA is blended into many functional foods and is sold worldwide including in Europe, the United States, and Asia. Despite the regulation of brain functions such as relaxation, sleep, and cognitive functions by dietary GABA intake, GABA has long been thought to be unable to cross the blood–brain barrier (BBB) [11,12,13]. Therefore, the mechanisms underlying the beneficial effects of GABA remain poorly understood.

The hypothalamus plays a crucial role in the control of the stress response, sleep, and feeding. Its activity is robustly regulated by peripheral information via two distinct routes, blood-mediated humoral pathways and sensory nerve-mediated neural pathways [14,15]. Vagal afferent neurons have their cell bodies in bilateral nodose ganglia (NGs) and bipolarly project to peripheral organs and the nucleus tractus solitarius (NTS)/the area postrema (AP) in the brainstem [16,17,18]. Meal-associated gastrointestinal and pancreatic hormones such as cholecystokinin, glucagon-like peptide-1, and insulin are elevated in circulation during the postprandial phase and are involved in inducing satiation or feeding suppression [14,15,19]. However, the transfer of these gastrointestinal or pancreatic hormones to the brain is tightly restricted by the BBB [14,15]. The branch terminals of vagal afferents sense these gastrointestinal or pancreatic hormones, thereby inducing satiation [14,15]. Therefore, dietary GABA, which is unable to cross the BBB, might act on the brain via vagal afferent neural pathways.

The present study aimed to clarify whether a single GABA administration by oral gavage affects brain function via vagal afferent nerves in mice. First, we examined changes in circulating GABA concentrations after peroral (po) GABA administration compared with its intraperitoneal (ip) administration. Next, we investigated the effects of po GABA injection on feeding behavior as an evaluation of brain function and examined the expression of phosphorylated extracellular signal-regulated kinase 1 and 2 (pERK1/2) in the NG, NTS, and AP. Phosphorylated ERK1/2 are known as cellular or neuronal activity markers because they are elevated by membrane depolarization and Ca^2+^ influx in PC12 cells [20], brain neurons [21,22,23], primary afferent (dorsal root ganglion) neurons [24], and nodose ganglion [19,25,26,27,28,29,30]. Furthermore, we examined whether the effects of GABA on feeding and pERK1/2 expression were counteracted by subdiaphragmatic vagotomy and the chemical denervation of capsaicin-sensitive sensory nerves. We found that po GABA administration immediately before refeeding suppressed food intake via vagal afferent nerves. However, the po administration of GABA alone did not change pERK1/2 expression in vagal afferent nerves, and po GABA 30 min before refeeding failed to decrease food intake. Therefore, we finally investigated the effect of GABA’s interaction with the diet on feeding regulation via vagal afferent nerves.

## 2. Materials and methods

### 2.1. Materials

GABA (>95% purity) was obtained from Pharma Foods International Co., Ltd. (Kyoto, Japan). Capsaicin (>93.0% purity, as capsaicinoids including capsaicin and dihydrocapsaicin), lithium chloride, and saccharin sodium dihydrate were purchased from Wako Pure Chemical Industries, Ltd. (Osaka, Japan). Liquid diet Ensure·H (EnsureH) was acquired from Abbott Japan (Tokyo, Japan).

### 2.2. Animals

Male C57BL/6J mice were purchased from the Japan Charles River Laboratory Japan (Yokohama, Japan) and housed for at least 1 week under controlled temperature (22.5 °C ± 2°C), humidity (55% ± 10%), and light (light phase; 7:30–19:30). Standard chow (CE-2, CLEA Japan, Tokyo, Japan) and water ad libitum were available to the mice. The animal experiments were carried out after receiving approval from the Institutional Animal Experiment Committee of the Kyoto Prefectural University and in accordance with the Institutional Regulations for Animal Experiments.

### 2.3. Measurement of Plasma GABA Concentration in the Postcaval Vein

Blood samples were collected from the postcaval vein of mice fasted for 5 h (9:00–14:00) under isoflurane anesthesia at 0, 15, 30, 60, 90, and 120 min after po (200 mg/kg, 10 mL/kg) or ip (20 or 200 mg/kg, 10 mL/kg) GABA administration, respectively. GABA was dissolved in saline. For oral administration, the solution was injected directly into the stomach using a stainless-steel feeding needle. The blood-sampling syringe contained heparinized saline (final concentration; 50 IU/mL). Plasma was collected after centrifugation (3000 rpm for 15 min at 4 °C) and stored at −80 °C until analysis. Then, plasma samples were mixed one–to–one with 5% 5-sulfosalicylic acid and then centrifuged at 12,500 rpm for 5 min. The supernatants were filtrated (0.45 µm) and transferred to HPLC vials. The GABA concentration in the plasma was analyzed using an LCMS-8045 mass spectrometer with a Prominence UFLC (Shimadzu Co., Kyoto, Japan) connected to an Intrada Amino Acid column (100 × 3 mm, Imtakt Co., Kyoto, Japan). The solvent system was a gradient of solvent A (acetonitrile/formic acid = 100/0.1) and solvent B (acetonitrile/100 mM ammonium formate = 20/80), the flow rate was 0.6 mL/min, and the following gradient was applied: 20% B in 0–4 min, 20%–100% B linear in 4–14 min, 100% B in 14–16 min. The injection volume was 5 µL for all samples, and the mass spectrometer was operated in positive mode. The amount of GABA in the samples (µg/mL plasma) was determined using a standard curve (GABA at 1.92, 9.6, 48, 240, and 1200 µg/mL).

### 2.4. Measurements of Food Intake and Locomotor Activities

Mice were housed in individual cages and sufficiently habituated to a standard powdered diet (CE-2, 3.4 kcal/g, CLEA Japan, Tokyo, Japan) in a feeding box (Shinano Manufacturing Co., Ltd., Tokyo, Japan) and to being handled for at least 1 week before experiments. Mice, including intact mice, capsaicin-treated mice, and vagotomized mice, were deprived of food from 18:00 with free access to water 1 day before the experiment (16 h fasting). On the next day at 9:50, GABA (2, 20, or 200 mg/kg, 10 mL/kg, dissolved in saline) or saline (10 mL/kg) was administrated po or ip, and CE-2 powdered diet was given at 10:00. The feeding box, including powder food and food spillage, was weighted at 1, 2, 3, 6, and 24 h after starting the test. The cumulative energy intake including food eaten (CE-2; 3.4 kcal/g) and administered GABA (3.95 kcal/g) at each time was expressed.

In another experimental protocol for food intake, on experiment day at 9:20, GABA (200 mg/kg, 10 or 40 mL/kg, dissolved in saline), liquid diet EnsureH (40 mL/kg), EnsureH + GABA (EnsureH, 40 mL/kg; GABA, 200 mg/kg), or saline (10 or 40 mL/kg) was administered by oral gavage to overnight-fasted mice. Then, 30 min after po injection, CE-2 powdered diet was provided at 10:00. EnsureH is a liquid diet that contains 1.5 kcal/mL with 14% of its metabolizable energy content as protein, 31.5% as fat, and 54.6% as carbohydrate.

Locomotor activity was estimated by the number of infrared beams broken in both X and Y directions using an activity monitoring system (ACTIMO-100N; Shinfactory, Fukuoka, Japan) combined with individual cages (size; W150 mm, D250 mm, H153 mm) with hanging feeding baskets. A pellet CE-2 diet was used to measure locomotor activity. GABA was perorally or intraperitoneally administered to overnight-fasted mice immediately before refeeding, and food intake and locomotor activity were concomitantly measured.

### 2.5. Conditioned Taste Aversion Test

The conditioned taste aversion test was performed, as previously reported [31]. To accustom mice to water deprivation schedules, they were allowed access to two water bottles for 2 h (10:00–12:00) for 5 days. On the 6th day, mice were given 0.15% saccharine instead of water for 0.5 h and were then injected with saline (10 mL/kg, po), GABA (200 mg/kg, po), or lithium chloride (LiCl, 3 nmol, 20 mL/kg, ip). The 7th day was the rest day, in which mice received free access to normal water for 2 h. On the 8th day, the 2-bottle preference (0.15% saccharine vs. water) test was performed for 0.5 h. Conditioned taste aversion was determined as a saccharine preference ratio: saccharine intake/total intake.

### 2.6. Systemic Capsaicin Treatment

To impair the capsaicin-sensitive sensory nerves, systemic capsaicin treatment was performed as described [26]. Mice were anesthetized with tribromoethanol (200 mg/kg, ip) followed by the subcutaneous (sc) administration of capsaicin at 50 mg/kg (5 mL/kg, solution composition: 10% ethanol, 10% Tween 80, and 80% saline). A second capsaicin (50 mg/kg, 5 mL/kg, sc) injection was performed 2 days later, following the same protocol. Finally, capsaicin (10 mg/kg, 10 mL/kg, ip) was injected into conscious mice 2 days later. A total of 5 to 10 days after the final treatment, feeding experiments (diet; powder CE-2) were performed. It has already been confirmed that the capsaicin-sensitive sensory nerve is denervated by this method [26]. After this experiment, to confirm whether the capsaicin treatment was successful, an eye-wiping behavioral test was performed by dropping capsaicin solution (0.5 mM in 5% DMSO, 10% Tween 80, and 85% saline), and a feeding test by ip injection of cholecystokinin-8 (4 µg/kg ip, vagal afferents dependent) was administered. In the capsaicin-treated mice, the number of eye-wipes significantly decreased (41.3 ± 2.33 times in intact mice vs. 10.3 ± 0.871 times in capsaicin-treated mice, *p* < 0.01 by unpaired *t*-test), and cholecystokinin-induced anorexigenic effects were abolished (0.453 ± 0.0368 g by saline ip vs. 0.380 ± 0.0863 g by CCK ip in capsaicin-treated mice, not significant).

### 2.7. Subdiaphragmatic Vagotomy

Bilateral subdiaphragmatic vagotomy was performed as described [31]. In brief, a midline incision was executed to achieve a wide exposure of the upper abdominal organs in mice anesthetized with tribromoethanol (200 mg/kg ip). The bilateral subdiaphragmatic trunks of vagal nerves along the esophagus were exposed and cut. In the sham operation group, these vagal trunks were exposed but not cut. Vagotomized and sham-operated mice were maintained on a nutritionally complete milk diet for human babies (Chilmil, Morinaga, Tokyo, Japan). One week after the operation, feeding experiments similar to the above were performed using a liquid diet (0.64 kcal/mL). Successful subdiaphragmatic vagotomy was confirmed by an increase in stomach weight, as previously reported [32].

### 2.8. Immunohistochemical Detection of pERK1/2 in the Nodose Ganglion, Medial Nucleus Tractus Solitarius, and Area Postrema

GABA (200 mg/kg, 10 or 40 mL/kg), EnsureH (40 mg/kg), EnsureH + GABA (EnsureH; 40 mL/kg, GABA; 200 mg/kg), or saline (10 or 40 mL/kg) was administered po at 10:00 to C57BL/6J mice fasted overnight (16 h). At 30 min after injection, mice were transcardially perfused with Zamboni’s solution (4% paraformaldehyde and 0.2 % picric acid in 0.1 M phosphate buffer at pH 7.4) under anesthesia. Bilateral NGs and brains were collected, postfixed in the same fixative for 2 and 4 h at 4 °C, respectively, and incubated in phosphate buffer containing 30% sucrose for 48 h at 4 °C.

Longitudinal sections (10 µm) of NGs were cut at 60 µm intervals using a precision cryostat (Leica Microsystems, Wetzlar, Germany). Coronal sections (40 µm) of the hindbrain were cut at 120 µm intervals using a freezing microtome (Yamato Kohki Industrial Co., Ltd., Saitama, Japan). Rabbit polyclonal antibodies against phospho-p44/42 MAPK (Thr202/Tyr204, pERK1/2) (1/500; #9101; Cell Signaling Technology, Danvers, MA, USA) and Alexa 488-conjugated goat antirabbit IgG (1/500; A11008; Thermo Fisher Scientific, Waltham, MA, USA) were used for the detection of pERK1/2 in NG and medial NTS/AP. Fluorescence images were acquired with an AxioObserver Z1 microscope and Axiocam 506 color camera (Zeiss, Oberkochen, Germany). Neurons immunoreactive to pERK1/2 in medial NTS (bregma −7.32 to −7.76 mm) were counted and averaged per section. In NGs, the number of pERK1/2-positive NG neurons in four sections per mouse was counted and averaged. The average pERK1/2-IR fluorescence intensity per unit area in the AP was analyzed using an imaging analysis system (NIH Image/ImageJ 1.50a).

### 2.9. Statistical Analysis

All data are shown as means ± SEM. Statistical analysis was performed with a two-tailed unpaired *t*-test or by one-way ANOVA. When ANOVA indicated significant differences among groups, those groups were compared using Dunnett’s or Tukey’s post hoc test. All statistical analyses were performed using Prism 7 (GraphPad Software, San Diego, CA, USA), and *p* < 0.05 was considered significant.

## 3. Results

### 3.1. The Ability to Increase Plasma GABA Concentration after po GABA Administration Is Markedly Lower Than That after Ip Administration

Researchers reported that orally administered GABA is absorbed into the blood in humans [8]. Here, we examined and compared the abilities to increase plasma GABA concentration after the po vs. ip administration of GABA. Ip injection of GABA at 200 mg/kg markedly increased plasma GABA concentration at 15 and 30 min after injection, peaking 15 min after (17,122 ± 1658 µg/mL at 15 min) and returning to baseline 60 min after (Figure 1). Additionally, po GABA administration into the stomach using a stainless feeding needle significantly elevated the plasma GABA concentration at 15 and 30 min after injection (7.26 ± 1.43 µg/mL at 0 min, 218 ± 39.9 µg/mL at 15 min, and 162.5 ± 41.4 µg/mL at 30 min, Figure 1). However, its peak value at 15 min after po injection was approximately 80-fold lower than that after ip administration (Figure 1). The potency of increasing plasma GABA concentration in po GABA administration at 200 mg/kg was similar to or slightly lower than that in ip GABA administration at 20 mg/kg (Figure 1). These results indicate that the oral ingestion of GABA limits its absorption or entry into the circulating blood.

### 3.2. Po GABA Administration Immediately before Refeeding Suppresses Food Intake without Aversive Behavior

Oral GABA supplementation reportedly regulates brain functions including suppressing stress and enhancing sleep [33]. Central neurons regulating stress and sleep, such as corticotropin-releasing hormone-containing neurons and orexin neurons, also control food intake. Therefore, to evaluate its regulation of brain function, we investigated the effect of oral GABA on food intake. In overnight-fasted mice, po administration of GABA at 20 and 200 mg/kg but not 2 mg/kg immediately before the start of refeeding slightly and significantly decreased food intake only during 0–0.5 h after injection (Figure 2A–C). These periodic feedings during 0–0.5 h in both GABA groups (20 and 200 mg/kg) were approximately 85% of those in the saline group (Figure 2B,C). Subsequently, cumulative food intake at 24 h after po GABA returned to normal levels, and body weight was not altered (data not shown). Several compounds exhibiting short-term feeding suppression such as oleoylethanolamide [34] reportedly induce locomotor impairment or aversion. We therefore examined the effects of po GABA 20 or 200 mg/kg on the locomotor activity and taste aversion. Po administration of GABA (20 and 200 mg/kg) immediately before refeeding reduced food intake during 0–0.5 h after injection but did not alter locomotor activity (Figure 2D). In the taste aversion test, po GABA (200 mg/kg) or po saline did not influence saccharine preference, unlike lithium chloride (Figure 2E). These results suggested that po GABA administration induced satiation without aversion or locomotor impairment.

Ip injection of GABA at 20 mg/kg demonstrated a similar capacity to increase plasma GABA to that of po GABA administration at 200 mg/kg (Figure 1). In contrast, ip injection of GABA at 20 mg/kg just before refeeding failed to decrease food intake (Figure 2F) and did not change locomotor activity (Figure 2F). Therefore, the increases in circulating GABA concentration after GABA administration might be not involved in GABA-induced feeding suppression.

### 3.3. Feeding Suppression by Dietary GABA Supplementation Is Attenuated by the Chemical and Surgical Denervation of Vagal Afferent Nerves

The vagal afferent nerves, which link several peripheral organs including the gastrointestinal tract and the brain, regulate food intake [14,15]. Therefore, we examined the participation of vagal afferents in the suppression of food intake by po GABA administration. The GABA effects (200 mg/kg, po) of suppressing cumulative food intake during 0.5–6 h after injection in control intact mice (Figure 3A) were abolished in mice pretreated with capsaicin subcutaneously (Figure 3B). Capsaicin is an agent that desensitizes the capsaicin-sensitive sensory nerves including vagal afferents [35]. Po GABA administration (200 mg/kg) immediately before refeeding significantly reduced cumulative milk-diet intake during 0.5–3 h in control mice receiving a sham operation (Figure 3C). In contrast, feeding suppression by GABA completely disappeared in subdiaphragmatic vagotomized mice (Figure 3D). These results demonstrated that intact vagal afferents are necessary for the anorexigenic effects of po GABA.

### 3.4. Po Administration of GABA Only without Feeding Does Not Alter Neural Activities in Vagal Afferent Neurons

We investigated whether a single GABA administration by oral gavage increases the expression of phosphorylated ERK1/2 (pERK1/2), as cellular/neuronal activity markers [19,24,25,26,27,28,29,30], in vagal afferent neurons. Po administration of GABA only at 200 mg/kg did not change pERK1/2 expression in the neurons of right and left NGs 30 min after injection (Figure 4A,B,G). Furthermore, po GABA did not alter the expression of pERK1/2 in the medial NTS and AP, which are the projection sites of the vagal afferents [16,17,18] (Figure 4C–F,H,I).

### 3.5. GABA-Supplemented Liquid Diet Enhances the Postprandial Activation of Vagal Afferent Neurons

Although dietary GABA alone did not alter the expression of phosphorylated ERK1/2 in vagal afferent neurons, GABA administration immediately before refeeding suppressed food intake via vagal afferents. Activation of a subclass of vagal afferents induces satiation or suppresses food intake [15,19,36]. Therefore, we suspected that a simultaneous stimulation with dietary GABA and a meal may induce vagal afferent activation and feeding suppression. According to Figure 1, most GABA administered po might be absorbed or metabolized from the gastrointestinal lumen within 30 min. The preadministration of 200 mg/kg GABA 30 min before refeeding did not alter food intake (Figure 5), contrasting with the effects of administration just before refeeding (Figure 2C and Figure 3A). These results indicate that immediately administering GABA before meals is essential for suppressing food intake, and the interaction between dietary GABA and meal-evoked factors might be important for feeding regulation mediated by vagal afferents.

Next, we investigated whether adding GABA to meals enhances the meal-evoked activation of vagal afferent nerves. Po administration of liquid diet EnsureH (40 mL/kg, approximately 850 µL and 1.3 kcal per mouse) to overnight-fasted mice, compared with saline (40 mL/kg), significantly increased the number of pERK1/2-positive neurons or pERK1/2-immunoreactive fluorescence intensity in bilateral NGs, NTS, and AP by approximately 1.5-fold (Figure 6A,B,D,E,G–I). These results indicate that meal intake activates the vagal afferent–brain axis. The coadministration of EnsureH and GABA significantly potentiated the meal-induced activation of the neurons in bilateral NGs, NTS, and AP, and these values were approximately 2–3 times those of the saline group and significantly higher than those of the EnsureH group (Figure 6A–I). Additionally, no difference was found in the activation of the left and right NGs by EnsureH + GABA (Figure 6G). These results indicate that the simultaneous intake of GABA and a meal markedly activates vagal afferents and innervating neurons.

### 3.6. The Coadministration of GABA and Liquid Diet Prevents Overeating via Capsaicin-Sensitive Sensory Nerves Including Vagal Afferents

Finally, we investigated whether the coadministration of GABA and a liquid diet suppresses food intake via sensory nerves including vagal afferents. Po GABA administration (200 mg/kg, 40 mL/kg) alone 30 min before refeeding, compared with saline (40 mL/kg, po), did not alter food intake in normal overnight-fasted mice (Figure 7A,B). In Figure 7A,B, control mice administered saline consumed food containing approximately 1.3 kcal for 0.5 h after refeeding (Figure 7A,B, white bar). Then, we preadministered EnsureH with approximately 1.3 kcal (40 mL/kg, approximately 850 µL) into overnight-fasted mice 30 min before refeeding and measured food intake sequentially. Unexpectedly, EnsureH preadministration alone did not change periodic food intake for 1–6 h after refeeding (Figure 7B), thereby inducing overeating until 6 h after refeeding (Figure 7A). This result indicates that meal intake by oral gavage exhibits a low potency to induce satiation. In contrast, the coadministration of EnsureH (40 mL/kg) and GABA (200 mg/kg) by oral gavage 30 min before refeeding remarkably suppressed food intake for 0–0.5 h after refeeding and thereby prevented overeating (Figure 7A,B). The feeding suppression by EnsureH + GABA was completely blunted in systemic capsaicin-treated mice (Figure 7C). These results suggest that the coadministration of GABA and a liquid diet potentiates meal-evoked satiation and prevents overeating via sensory nerves including vagal afferents.

## 4. Discussion

Dietary GABA demonstrates beneficial effects on brain function, such as relieving anxiety and improving sleep quality [33]. Since GABA reportedly cannot cross the BBB [11,12,13], its underlying mechanisms remain poorly understood. Here, we found that a single po GABA administration at 200 mg/kg suppressed short-term food intake without aversive behaviors in mice. The increase in circulating GABA concentration after po GABA administration at 200 mg/kg was similar to that after ip GABA injection at 20 mg/kg, which did not change food intake. Therefore, these results indicate that the elevation of blood GABA concentration is not involved in feeding regulation by dietary GABA. The suppression of feeding by po GABA was blunted by the chemical denervation of capsaicin-sensitive sensory nerves and surgical subdiaphragmatic vagotomy. However, po administration of GABA alone did not alter pERK1/2 expression as cellular/neural activity markers in vagal afferents and innervating brain sites including NTS and AP. Furthermore, we found that po GABA immediately before refeeding suppressed food intake, but po GABA 30 min before refeeding failed to decrease food intake. These results suggest that dietary GABA interacts with meal-evoked satiation and the postprandial activation of vagal afferents. We found that the coadministration of liquid diet and GABA potentiated the postprandial activation of vagal afferent neurons and enhanced meal-evoked satiation, thereby preventing hyperphagia. Therefore, dietary GABA might cooperate with meal-evoked factors such as nutrient and postprandial hormones to interact with vagal afferents, being relayed to brain function including food intake suppression.

The ability to increase plasma GABA concentration after po GABA administration was markedly lower than that after ip administration. However, the absorption and metabolism of dietary GABA in the intestine have not been completely elucidated, whereas other amino acids such as glutamate and aspartate in the diet are mostly catabolized by the small intestinal mucosa, and their concentrations in circulating blood do not increase markedly [37,38]. Therefore, circulating GABA concentrations do not markedly increase by oral GABA intake, as most of it may be metabolized in the intestine and liver. GABA is widely used as a supplement and medicine (GAMMALON Tablets 250 mg, Alfresa Pharma Co., Ltd., Japan). No adverse effects of oral GABA (~5 g/day) ingestion have been observed [10,39,40,41]. Conversely, muscimol, which is a potent GABA_A_ receptor agonist, is known as a psychoactive and toxic ingredient in the mushroom *Amanita muscaria* that crosses the BBB [42]. Therefore, the high safety of dietary GABA may be because it does not cross the BBB and because its blood levels are not easily elevated after its oral ingestion.

We found that a single po administration of GABA immediately before refeeding induces satiation and suppresses food intake via the vagal afferent nerves. However, the mechanism by which GABA acts on the vagal afferents remains unproved. GABA is a major inhibitory neurotransmitter in the brain acting through either chloride-permeable ionotropic GABA_A_ or metabotropic GABA_B_ receptors [43]. A quick search in the mice vagal afferent atlas by single-cell RNA sequencing transcriptomic analysis from Kupari et al. [44] seems to indicate that genes coding for GABA_A_ and GABA_B_ receptor subunits are expressed in all or a subclass of these neurons. Previous experiments using ferrets showed that some of the vagal afferent neurons expressed immunoreactivity for the GABA_B_ receptor, which is involved in the inhibition of gastric mechanical sensory activity [45]. On the other hand, other studies reported that GABA depolarized the desheathed nerves isolated from NGs via GABA_A_ receptor using in vitro electrophysiology [46,47]. Hence, these previous findings of GABA action on the vagal afferents using different models and species do not provide a unified explanation. In the present study, we demonstrated that a single po administration of 200 mg/kg GABA at doses that suppress food intake did not increase pERK1/2 expression in NG neurons in vivo. Furthermore, po administration of GABA alone 30 min before refeeding failed to suppress food intake, whereas the combination of liquid diet and GABA significantly increased pERK1/2 expression in NG neurons and prevented overeating due to feeding suppression. These results suggest that po GABA administration might act indirectly, but not directly, on the vagal afferent nerves. We suspect that it might interact with meals or meal-evoked factors to enhance the postprandial activation of vagal afferents. Many gastrointestinal and pancreatic hormones such as cholecystokinin, glucagon-like peptide-1 (GLP-1), peptide YY, and insulin directly activate vagal afferent neurons and induce satiation [14,15,19]. GLP-1 released by meals and food ingredients, such as the rare sugar D-allulose, activate a subclass of vagal afferent neurons expressing GLP-1 receptor, thereby inducing satiation, enhancing insulin secretion, and regulating glucose metabolism [19,48]. GABA reportedly stimulated GLP-1 release from the GLUTag cell line in an in vitro situation [49]. In vivo, simultaneous po administration of liquid meal and GABA, but not GABA alone, robustly increased pERK1/2 expression as cellular/neural activity markers in vagal afferent neurons. GLP-1 is rapidly degraded to inactive form by dipeptidyl peptidase-4 (DPP-4), resulting in a short half-life of 1–2 min [50]. Therefore, it is possible that GABA may enhance GLP-1 action on the vagal afferents by inhibiting DPP4 activity. The mechanism by which dietary GABA interacts with vagal afferent nerves ought to be examined in future research.

Vagus nerve stimulation, which is a form of neuromodulation that utilizes an implantable pulse generator, recently became an FDA-approved treatment for brain dysfunctions such as epilepsy, depression, and obesity [15,49,51]. Additionally, dietary GABA regulates brain function through its stress-reducing and sleep-enhancing activities. However, its underlying mechanisms remain undemonstrated because GABA has long been thought to not cross the BBB [33]. The present study demonstrates that dietary GABA activates the vagal afferent nerves in collaboration with meal-evoked factors and regulates brain function including feeding behavior. In other words, dietary GABA enhances the postprandial activation of vagal afferents, thereby potentiating postprandial satiation. Recent growing evidence has indicated that gut–brain signal-related vagal afferents and gut microbiota influence emotional behaviors [52,53,54,55]; thus, dietary GABA might act on the gut–brain axis to treat stress-related disorders. By clarifying the detailed mechanism of vagal afferent activation by dietary GABA, the control of vagal nerve activity by dietary GABA might become a useful tool for improving brain functions including feeding, metabolism, and other mental functions.

## Figures and Tables

**Figure 1 nutrients-14-02492-f001:**
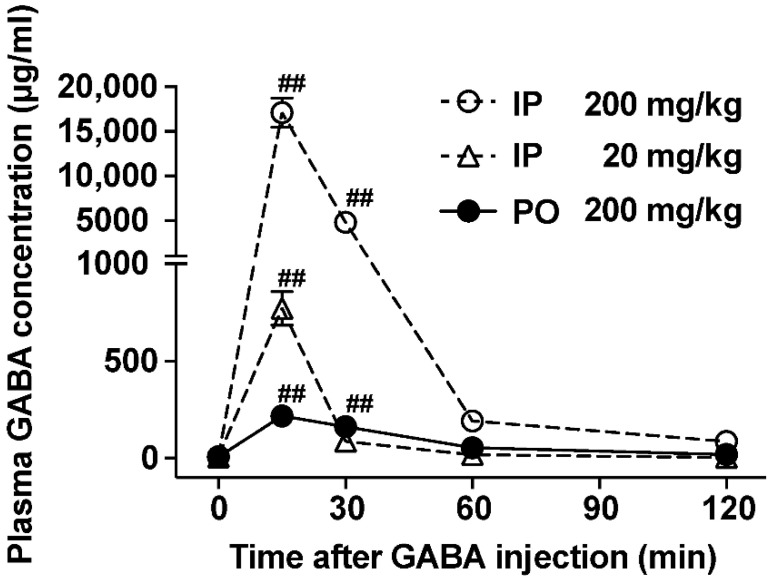
Plasma GABA concentrations in mice administered with peroral (po) vs. intraperitoneal (ip) GABA. Time course of GABA concentrations in the postcaval vein plasma after po administration of GABA (200 mg/kg; closed circle) or ip injection of GABA (200 mg/kg; open circle, 20 mg/kg; open triangle). *n* = 4–6. ^##^ *p* < 0.01 by one-way ANOVA followed by Dunnett’s test vs. 0 min in each group. GABA: Gamma-aminobutyric acid.

**Figure 2 nutrients-14-02492-f002:**
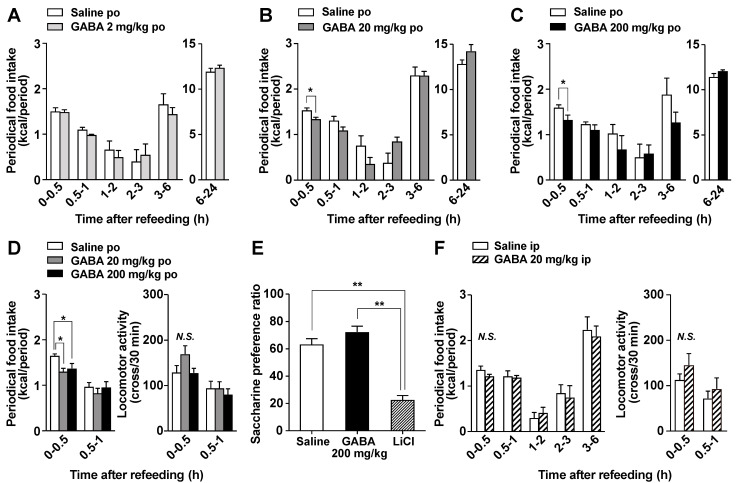
Po GABA administration immediately before the start of refeeding suppresses food intake in a dose-dependent manner without aversive behaviors. (**A**–**C**): Periodic food intake in response to the po administration of GABA at 2 (**A**), 20 (**B**), or 200 mg/kg (**C**) or saline immediately before refeeding in mice fasted overnight (16 h). Periodic food intake (kcal) for 0–0.5 h after GABA injection includes the energy from eaten chow and injected GABA. The amount of energy in GABA was calculated as 3.95 kcal/g. Therefore, the energy GABA administered at 2, 20, and 200 mg/kg to the mice was 0.163 ± 0.00473 (**A**), 1.69 ± 0.0392 (**B**), and 15.8 ± 0.219 (**C**) cal per mouse, respectively. * *p* < 0.05 by unpaired *t*-test. (**D**): Periodic food intake (left) and locomotor activity (right) in response to po administration of GABA 20 or 200 mg/kg immediately before refeeding in mice fasted overnight (16 h). * *p* < 0.05 by one-way ANOVA followed by Tukey’s test. *n* = 7. (**E**): In the taste aversion test, saccharine preference was measured 2 days after the injection of saline (po), GABA (200 mg/kg, po), or lithium chloride (LiCl, 3 mmol/kg, ip). ** *p* < 0.01 by one-way ANOVA followed by Tukey’s test. *n* = 6–12. (**F**): Ip injection of GABA at 20 mg/kg immediately before refeeding did not change food intake (left) or locomotor activity (right) in overnight-fasted mice. *n* = 6. GABA: Gamma-aminobutyric acid. N.S.: not significant.

**Figure 3 nutrients-14-02492-f003:**
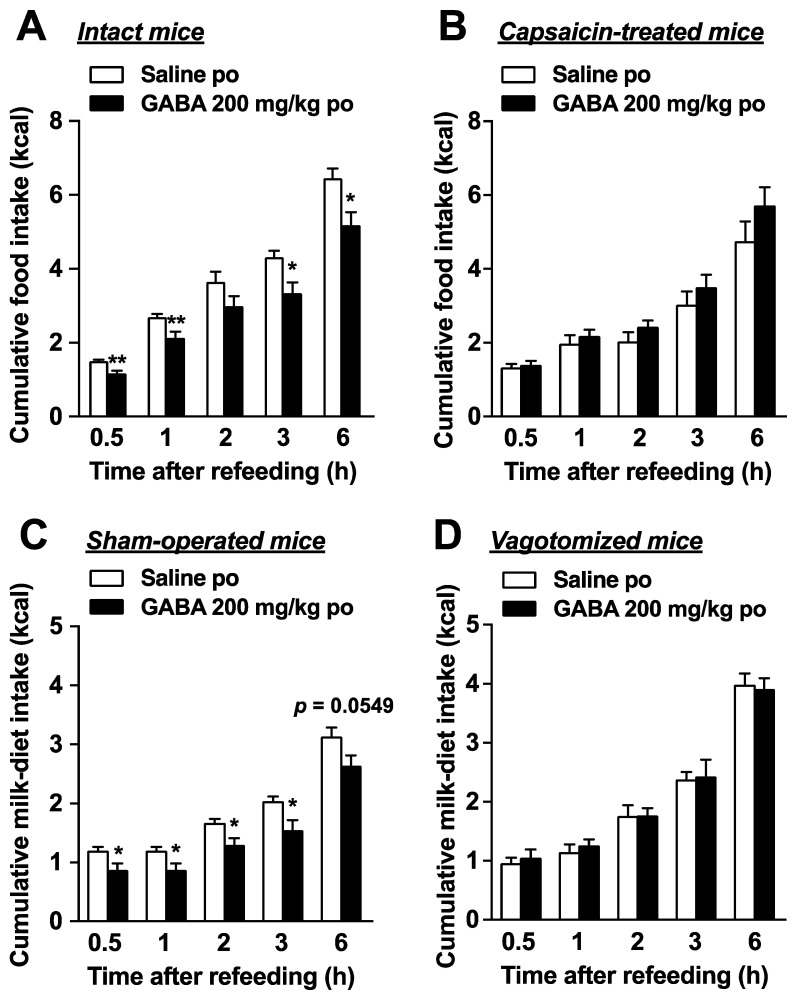
Peroral GABA administration suppresses food intake via vagal afferents. (**A**): Peroral administration of GABA (200 mg/kg) immediately before refeeding significantly reduced food intake in intact mice fasted overnight (16 h). *n* = 6. (**B**): The feeding suppression by po GABA administration (200 mg/kg) immediately before refeeding was abolished in mice whose sensory nerves including vagal afferent nerves were denervated by systemic treatment with capsaicin. *n* = 6. (**C**,**D**): GABA (200 mg/kg, po) reduced cumulative milk-diet intake in sham-operated (**C**) but not subdiaphragmatic vagotomized mice (**D**). *n* = 5–8. * *p* < 0.05, ** *p* < 0.01 by unpaired *t*-test. po, peroral. GABA: Gamma-aminobutyric acid.

**Figure 4 nutrients-14-02492-f004:**
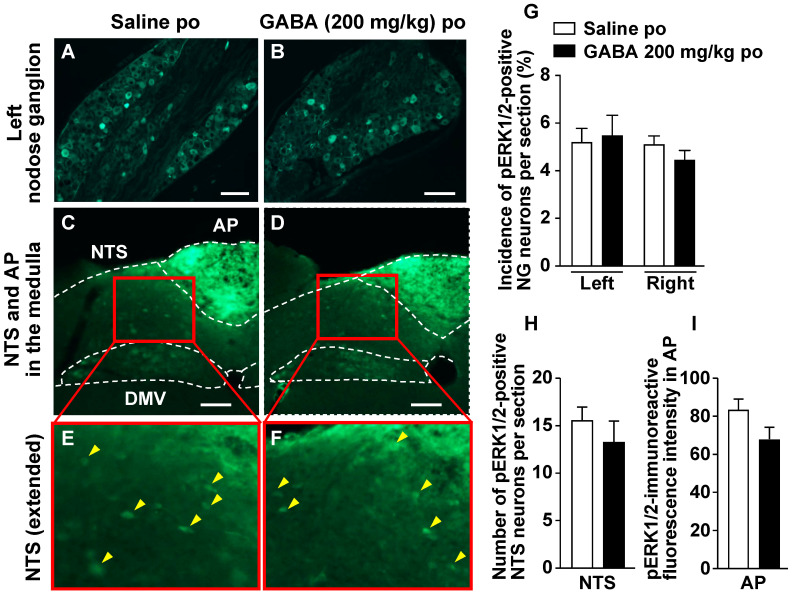
A single po administration of GABA does not alter ERK1/2 phosphorylation in the nodose ganglia (NGs), nucleus tractus solitarius (NTS), or area postrema (AP). *A–D*: Left NG (**A**,**B**) and medial NTS and AP in the medulla (**C**,**D**) sections immunostained for pERK1/2 30 min after po administration of 200 mg/kg GABA or saline to overnight-fasted mice. (**E**,**F**): Extended pictures of red squares in (**C**) to (**D**), respectively. Yellow arrowheads indicate pERK1/2-positive neurons. Scale bar, 100 µm. (**G**–**I**): The incidence of pERK1/2-positive neurons in the left and right NG neurons (**G**) and the number of them in the bilateral NTS neurons (**H**). The fluorescence intensity of pERK1/2-immunoreactivity per unit area in AP is indicated in I. *n* = 4. DMV, dorsal motor nucleus of vagus nerves. po, peroral. GABA: Gamma-aminobutyric acid. Dotted line: borderline indicating brain region.

**Figure 5 nutrients-14-02492-f005:**
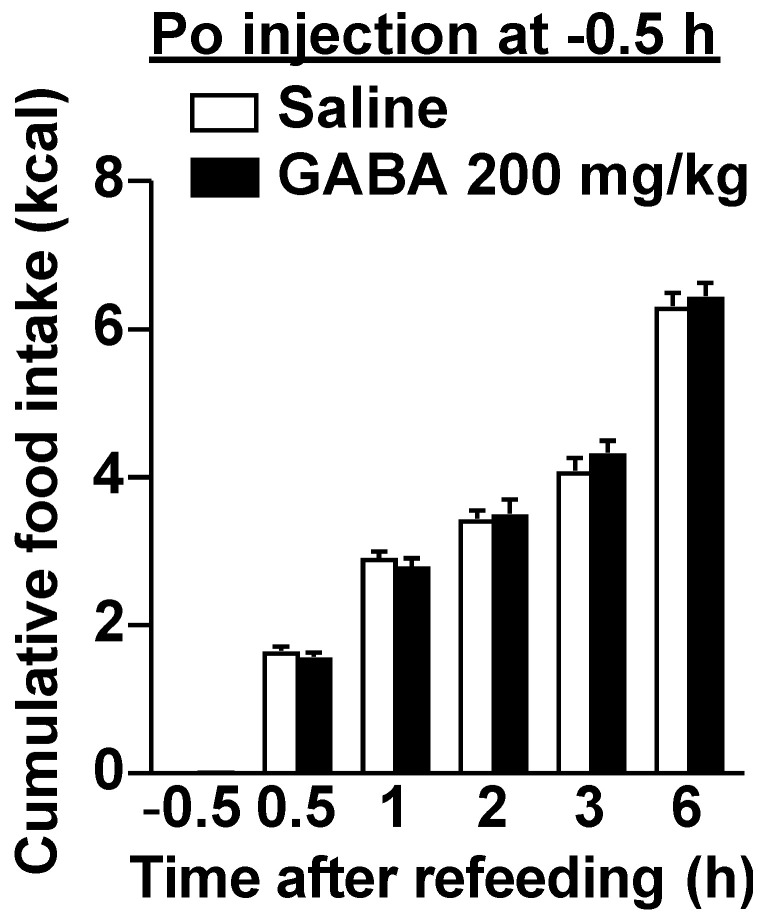
The preadministration of GABA 30 min before refeeding did not alter food intake. A single po administration of GABA at 200 mg/kg 30 min before refeeding failed to decrease food intake in overnight-fasted mice. *n* = 12. At −0.5 h, the value of saline group is 0 and that of GABA is 0.0174 ± 0.000278 kcal, which are the energy values of the injected solution. po, peroral. GABA: Gamma-aminobutyric acid.

**Figure 6 nutrients-14-02492-f006:**
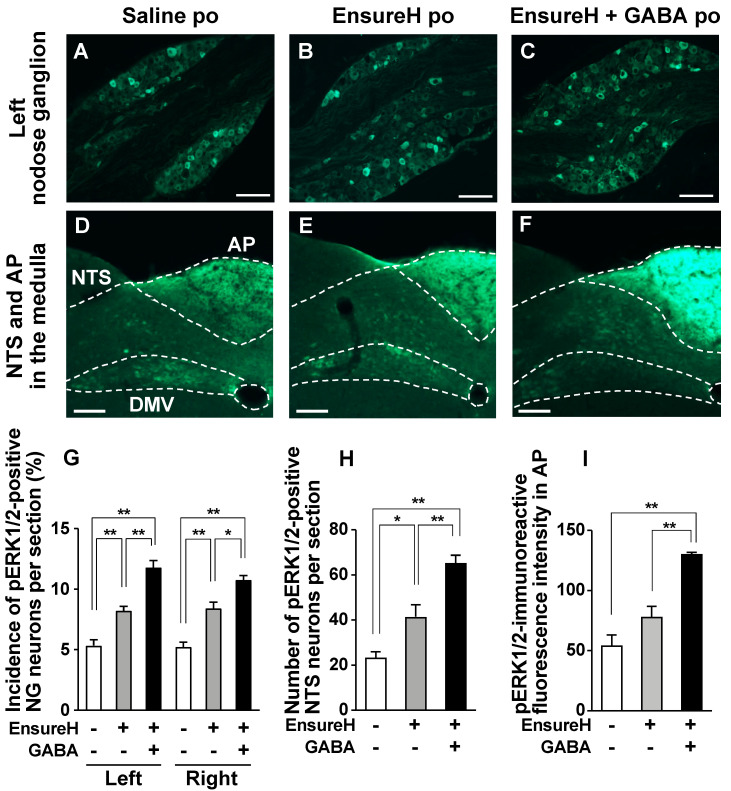
The coadministration of liquid diet EnsureH and GABA potentiates the postprandial activation of neurons in nodose ganglia (NGs), nucleus tractus solitarius (NTS), and area postrema (AP). (**A**–**F**): Left NG (**A**–**C**) and medial NTS and AP in the medulla (**D**–**F**) sections immunostained for ERK1/2 phosphorylation 30 min after po administration of EnsureH (40 mL/kg), EnsureH + GABA (40 mL/kg and 200 mg/kg), or saline (40 mL/kg) in overnight-fasted mice. Scale bar, 100 µm. (**G**–**I**): The incidence of pERK1/2-positive neurons in the left and right NG neurons (**G**) and the number of them in the bilateral NTS neurons (**H**). The fluorescence intensity of pERK1/2-immunoreactivity per unit area in AP is indicated in I. * *p* < 0.05, ** *p* < 0.01 by one-way ANOVA followed by Tukey’s test. *n* = 4. DMV, dorsal motor nucleus of vagus nerves. po, peroral. GABA: Gamma-aminobutyric acid. Dotted line: borderline indicating brain region. EnsureH+, po administration of EnsureH (gray columns). GABA+, po administration of EnsureH containing GABA (black columns). EnsureH– and GABA–, po administration of saline (white columns).

**Figure 7 nutrients-14-02492-f007:**
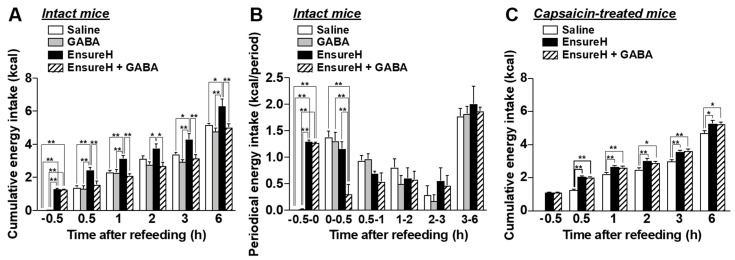
The coadministration of liquid diet EnsureH and GABA markedly prevents overeating through sensory nerves including vagal afferents. (**A**,**B**): Po administration of only GABA (200 mg/kg, 40 mL/kg) 30 min before refeeding, compared with saline (40 mL/kg, po), did not alter food intake in intact mice fasted overnight. The preadministration of only EnsureH (40 mL/kg) by oral gavage 30 min before refeeding did not alter food intake (**B**), thereby increasing cumulative food intake and inducing overeating until 6 h after refeeding (**A**). The coadministration of EnsureH (40 mL/kg) and GABA (200 mg/kg) markedly suppressed food intake for 0–0.5 h after refeeding (**B**) and thereby prevented overeating (**A**). *n* = 5–6. (**C**): In capsaicin-treated mice, the simultaneous administration of EnsureH (40 mL/kg) and GABA (200 mg/kg) failed to decrease food intake. *n* = 11. * *p* < 0.05, ** *p* < 0.01 by one-way ANOVA followed by Tukey’s test. GABA: Gamma-aminobutyric acid.

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
