# Peer review of "Dietary Gamma-Aminobutyric Acid (GABA) Induces Satiation by Enhancing the Postprandial Activation of Vagal Afferent Nerves"

_nutrients, 2022, doi:10.3390/nu14122492_

Round 1

Reviewer 1 Report

In their manuscript, Nakamura and colleagues investigate whether oral administration of GABA is anorexigenic in male mice. This question is original and highly relevant, as GABA is naturally present in foods and also added to many nutritional supplements. 

The authors conclude that peroral adminstration of GABA decreases short-term food intake via a modulation of vagal afferent neurons activity. Concerning this conclusion, I have major questions that preclude publication of the manuscript in its current form.

In addition, the manuscript leaves many questions unanswered. Notably, the authors have not elucidated how GABA interacts with vagal afferents nor have they investigated the central mechanisms that could trigger the anorexigenic effects of oral GABA.

MAJOR

- First, across many experiments, GABA reduces food intake only when administered "immediately" before food access, and only for a short period of time (<30 minutes).
In my experience, this is rather the case of compounds that reduce food intake in an unspecific manner rather than the case of true satiation compounds. Hence, the authors should respond to the following subquestions.
Does the decrease in food intake after po GABA occurs via a delay of the first meal (increase in latency to eat) or rather a decrease in first meal size?
Could the authors report the behavior of the mice right after injection? I understand that the authors have excluded aversion as a potential "unspecific" mechanism, but some compounds modify the immediate behavior of the animals post injection (via a reduced locomotor activity, light nausea or increase in anxiety-like behavior...) without inducing CTA. This is the case for example of oleylethanolamide (PMID 29388342). For example, a measurement of mice locomotor activity or pica would give more credit to the possibility that po GABA is truly anorexigenic.

- What was the rationale to exclude female mice from the study?

- IP injection should give a compound access to vagal afferent terminals in the gut. IP injections increase for example the GLP1 concentration in intestinal lymph (which is what vagal afferent terminals in the gut are exposed too), see PMID 29341824. 
So, if exogenous GABA decreases food intake via vagal afferents, it is unclear why IP GABA does not do so. Could you attempt to explain this? Related to the first question, is the behavior of mice different when injected ip vs after po GABA?

- The rationale of using pERK as a general activation marker in the nodose ganglia is unclear. The two articles that are cited refer to DRG neurons or to the response to glucagon administration. 
This does not mean that the absence of an increase in pERK after GABA adminstration equates absence of activation of vagal afferents. 
If pERK is validated as an immediate early gene of vagal afferent neurons, this should be explained. If not, other IEG might be tested before a conclusion can be drawn

- The overall idea that GABA can modulate vagal afferent neurons activity would benefit from the identification of GABA receptors on these neurons.
A quick search in the mice vagal afferent atlas from Kupari et al. (https://ernforsgroup.shinyapps.io/vagalsensoryneurons/) seems to indicate that genes coding for GABAa and GABAb receptor subunits are expressed in quite a high level in all or a subset of these neurons.
This is however, just mining available ScRNA seq data. Can the authors demonstrate the presence of GABA receptors directly in mice vagal neurons (IHC; RNAscope)? 

MINOR

- Related to FIg 1: While it is clear that the aim was to test whether oral GABA is found in the blood, the rationale for using IP as positive control is unclear. IV would be a more direct control. 
Also, technically, a compound could activate vagal afferents neurons in the intestinal mucosa without being found in the blood, so it is hard to conclude anything from this experiment, see PMID 29341824 again.

- Since the 20 mg/kg dose also reduces FI, could GABA be detected in the blood at this dose?

- Fig1AB: I assume that PO line is the same on both graphs? Maybe put all three lines on one graph with a broken y axis or a log scale?

- Fig 2: It is unclear why graphs for periodical food intake are splitted by dose, while the cumulative graph shows all doses together. For ease of use, I would suggest to choose either to split or to group, but not both (especially for two variables like periodical/cumulative FI that are technically the same)

- Fig 3A: Spontaneously, based on the 2 by 2 presentation of the figure, I thought that this subpanel was the control for the capsaicin treated mice. I would suggest to place this subpanel with the previous figure, if possible.

- Was there a validation procedure for capsaicin treatment? where are the controls for capsaicin treated mice?

- L 256-7: this should be rephrased. You do not demonstrate an interaction. Stricto sensu, you  show that intact vagal afferents are necessary for the anorexigenic effects of po GABA.

- Fig 4: as po GABA does not decrease FI when administered 30 min before food, why did you measure pERK at 30 min? It is likely that both food intake effects and ERK phosphorylation are gone after 30 min.
Repeat the experiment after 15 min?

- Fig 5A-C: the number of neurons per section seem to vary. Could you normalize by the number of neurons per section?

- Fig 5F: AP is higly overexposed: I don't think it is reasonable to quantify anything from this. Adjust if possible. 

- FIg 7: I am not sure that a summary figure is helpful here. It mostly highlights that you have neither elucidated how GABA interacts with vagal afferents nor the central mechanisms associated.

Author Response

Blue characters in the letter mean our responses.

Reviewer 2 Report

To authors

11)     Lines 200- 202: “The potency of increasing plasma GABA concentration in po GABA administration at 200 mg/kg was similar to or slightly higher than that in ip GABA administration at 20 mg/kg (Figure 1B)”.

Are you sure it's not the other way around? I think she is still the shortest.

22)     The authors indicate that elevated blood GABA concentration is not involved in the regulation of dietary GABA feeding. Suppression of feeding by oral injection of GABA (po) was mitigated with chemical denervation of sensory nerves by capsaicin and subdiaphragmatic vagotomy. In addition, the authors determine that po GABA before refeeding suppresses food intake, however, po GABA 30 minutes before refeeding does not suppress food intake. But with the liquid diet Ensurance H increases hypernutrition and with dietary GABA it enhances vagal afferent activity.

I agree that these results are interesting in suggesting that dietary GABA interacts with some hedonic factor through hormonal action, from the food itself, or from the postprandial. As you indicate in figure 7, dietary GABA does not act directly on vagal afferents, but cooperates with unknown mechanisms that could be endocrine factors of hedonic control (satiation) that will activate vagal afferent nerves in the background, and induce satiety. Within these mechanisms or molecular pathways that could be intermediaries, several gastrointestinal and pancreatic hormones capable of acting on parasympathetic afferenet neurons are named, emphasizing a greater role in GLP-1. However, GLP-1 is a substrate for an enzyme that regulates metabolic and energy control, I would also recommend monitoring the status of Dipeptidyl peptidase IV (DPP4) activity.

It would be interesting if the authors defined at the end of the discussion a possible study path to follow in future experiments.

33)     There are a lot of literature which explain that abdominal vagal afferents in rats display anhedonic behavior and increase behavioral despair. Accumulating evidence demonstrates that the gut-brain axis, which is primarily regulated by the vagus nerve, is involved in stress, suggesting a communication between the “gut-vagus-brain” pathway and the GABAergic neuronal system. Interestingly, growing evidence has shown that gut-brain signals influence emotional behaviors, and the gut-brain axis may be a possible target for treating stress-related disorders. It suggests that vagal afferents may connect with the limbic system and affect the GABAergic system in the CNS. Authors could looking for a possible explanation in these route.

44)     But on the other hand, the authors do not consider other new aspects that should be considered. Recent reviews has summarized the psychophysiological effects of prebiotics and discussed the important roles of bacteria-gut-brain signals in psychobiotic activity. In addition to neurotransmitters and neuropeptides, the gut microbiome, are also involved in the gut-brain communications. Together could be involved in the central nervous system (CNS) and involved in regulating the function of the digestive system. Authors could consider if the liquid diet (Ensurance H) could modulate early microbiome changes and to modify energetic metabolism and satiety. 

Author Response

(The authors gave the same response as above.)

Round 2

Reviewer 1 Report

I thank the authors for their thorough answers to my previous comments.

Some of my comments remain unanswered due to the short time that was given to the authors for revision (major comments 1.1 and 5, notably), but these issues do not preclude publication.

I would like, however, to pinpoint the fact that the authors’ comment about the exclusion of female mice from the study is really outdated. Decribing the male mice as the “more simple” model, while considering variations in the estrous cycle as a biological annoyance to be eliminated is conceptually preoccupying. I would encourage the authors to systematically include both sexes in future studies: this is actualy a chance for your research, as coherent results between sexes increases the robustness of the conclusion, while discrepancies between sexes is often an excellent way to get to the underlying mechanisms (so only advantages).